# HOUGH VOTING-BASED PROMPT LEARNING FOR SEGMENT ANYTHING MODEL

## ABSTRACT

Segment Anything Models (SAMs) like SEEM and SAM have achieved great performance on various downstream datasets at the cost of crafting spatial and semantic prompts. Previous prompt learning methods can learn prompts automatically but largely focus on learning semantic prompts, while how to learn effective spatial prompts that are important to SAMs is largely under-explored. Inspired by Hough Voting that detects a complex object by voting from its parts, we propose Hough Voting-based Spatial Prompt Learning (HoughSpaPL) that designs three types of voting mechanisms to learn three distinct spatial prompts for different sub-regions of the visual concept (e.g., things and stuff), which capture complementary spatial clues and vote together to guide SAMs to generate a precise segmentation mask for the visual concept. Following the same philosophy, we design Hough Voting-based Semantic Prompt Learning (HoughSemPL) that learns distinct semantic prompts for different sub-regions of the visual concept, which capture complementary semantic clues and vote together to predict a accurate semantic label for the generated mask. Extensive experiments show that our proposed techniques achieve superior prompt learning performance over popular segmentation datasets. Codes will be released.

## 1 INTRODUCTION

Segment Anything Models (SAMs), like Segment Everything Everywhere Model (SEEM) Zou et al. (2023) and Segment Anything Model (SAM) Kirillov et al. (2023), have recently shown great zero-shot generalization performance in image segmentation across various downstream tasks and datasets Cordts et al. (2016); Zhou et al. (2017). The remarkable success of SAMs are largely attributed to the novel "Promptable Segmentation" framework, in which the model takes handcrafted prompts as inputs and predicts the expected segmentation masks accordingly. Specifically, promptable segmentation usually involves two categories of prompts comprising (1) spatial prompts such as points, and (2) semantic prompts like free-form texts, where the former provides spatial clues while the latter provides semantic information which together guide the model to segment and classify the expected visual concepts.

On the other hand, manually crafting proper prompts for each downstream task and dataset not only requires domain expertise Zhou et al. (2022b;a); Liu et al. (2021) but also is often tedious and time-consuming Zhou et al. (2022b;a); Parisot et al. (2023). To tackle this issue, prompt learning (Zhou et al., 2022b;a; Liu et al., 2021) has been extensively studied, which learns effective prompts by optimizing learnable prompt vectors with few-shot data, such as CoOp (Zhou et al., 2022b), LOCN (Parisot et al., 2023) and PLOT (Chen et al., 2022). However, previous prompt learning methods Zhou et al. (2022b); Parisot et al. (2023); Chen et al. (2022) largely focus on learning semantic prompts that provide semantic clues for category classification, while how to learn effective spatial prompts that could provide spatial clues to guide SAMs to locate and segment the expected visual concepts accurately is largely under-explored.

Inspired by Hough Voting Ballard (1981); Woodford et al. (2014); Qi et al. (2019); Samet et al. (2022) that detects a complex object based on the voting from its parts, we propose to learns distinct spatial prompts for different sub-regions of the visual concept (e.g., things and stuff), which capture complementary spatial clues and vote together to guide SAMs to generate a precise segmentation mask for the visual concept, as shown Figure 1. To this end, we propose Hough Voting-based Prompt

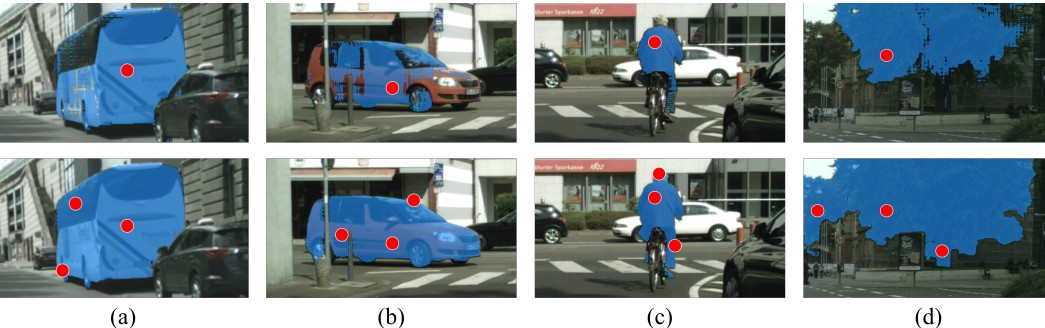

(a)  (b)  (c)  (d)

Figure 1: **Comparison of traditional Spatial Prompt Learning (SpaPL) and our HoughSpaPL.** Traditional SpaPL Huang et al. (2024) learns a single spatial prompt for each visual concept (e.g., thing or stuff), which struggles when handling complex visual concepts with irregular boundaries. Inspired by Hough voting, we propose HoughSpaPL that learns distinct spatial prompts that focus on different sub-regions of the visual concept (i.e., the inner, middle, and outer regions), capturing complementary spatial clues (e.g., information nearing the center and around the boundary) and voting together to guide SAMs to generate a precise segmentation mask.

Learning (HoughPL) that consists of Hough Voting-based Spatial Prompt Learning (HoughSpaPL) and Hough Voting-based Semantic Prompt Learning (HoughSemPL).

HoughSpaPL designs three types of voting mechanisms including *inner-region voting*, *middle-region voting* and *outer-region voting*, which encourage to learn three spatial prompts with different spatial focuses that provide complementary spatial clues and vote together to generate a precise segmentation mask. Specifically, *inner-region voting* enforces the prompt to focus on and learn from the inner-region of object (e.g., the region near object center), and then predicts a segmentation mask for a whole object by voting from the information learnt from the inner-region only. Similarly, *middle-region voting* enforces the prompt to learn and vote from the middle-region of object (e.g., the region around object boundary) a segmentation mask for a whole object. And *outer-region voting* enforces the prompt to learn and vote from the outer-region of object (e.g., the region outside object that may contain other objects and/or background) a segmentation mask for a whole object. Then, HoughSpaPL[1] generates the final segmentation mask by fusing the three masks predicted from these three prompts with associated voting mechanisms.

HoughSpaPL has two desirable features for spatial prompt learning for SAMs: (1) it learns different spatial prompts that capture complementary spatial clues, e.g., inner-region information can facilitate locating the target thing/stuff while middle-region and outer-region information could help predict a precise segmentation boundary, leading to a robust segmentation; (2) its voting mechanisms enable and encourage information flow among different regions, i.e., from inner, middle and outer regions towards the whole object, learning rich contextual information and benefiting image segmentation.

On the other hand, HoughSemPL learns distinct semantic prompts for different sub-regions of the visual concept, which capture complementary semantic clues and vote together to predict a accurate semantic label for the generated mask. Specifically, HoughSemPL learns three different types of semantic prompts for the three spatial prompts in HoughSpaPL respectively. As the three spatial prompts in HoughSpaPL are enforced to focus on and learn from different regions, the three corresponding semantic prompts will be encouraged to focus on different regions as well and thus learn complementary semantic information. For example, as shown in Figure 1 (a), the three spatial prompts focus on the regions of bus hood, side window, and wheel, respectively, from where the three corresponding semantic prompts learn complementary semantic information.

The contributions of this work can be summarized in three major aspects. First, we design HoughPL, a new prompt learning method for SAMs, which explores the idea of Hough voting to learn complementary spatial and semantic prompts and vote together to guide SAMs to segment. Second, HoughPL introduces HoughSpaPL and HoughSemPL. HoughSpaPL designs three types of voting mechanisms which encourage to learn three spatial prompts with different spatial focus and vote

---

[1]Note our HoughSpaPL works for both things and stuff (i.e., both object and background visual concepts) in the same manner. Here we only mention 'object' for simplified illustration.

together to prompt SAMs. Similarly, HoughSemPL learn three different types of semantic prompts to guide SAMs. Third, extensive experiments demonstrate state-of-the-art performances of the proposed HoughPL over multiple widely adopted segmentation datasets.

## 2 RELATED WORKS

**Segmentation Anything Models (SAMs)** have achieved impressive zero-shot generalization and interactive segmentation performance over a wide range of datasets. SAMs such as SAM Kirillov et al. (2023), SEEM Zou et al. (2023), and Semantic-SAM Li et al. (2023) represent significant advances in image segmentation by introducing a promptable architecture. They process handcrafted spatial prompts (i.e., points, boxes, or masks) and/or semantic prompts (i.e., textual descriptions) to generate corresponding segmentation masks, facilitating the application in various scenarios. Fast-SAM Zhao et al. (2023) enhances inference speed, while HQ-SAM Ke et al. (2024) improves segmentation quality. However, acquiring effective prompts for SAMs is a non-trivial problem that is largely under-explored. We focus on learning effective prompts that improves the SAMs' performance on task-specific or domain-specific downstream datasets with limited labeled samples.

**Prompt Learning** aims to transfer a foundation model towards downstream domain by optimizing prompts with limited labeled data. It has been extensively researched in NLP foundation models Zhong et al. (2021); Li & Liang (2021); Lester et al. (2021); Liu et al. (2023) and image classification foundation models (CFMs) Zhou et al. (2022b); Parisot et al. (2023); Zang et al. (2022), but is still in its early stages Huang et al. (2024) in SAMs. For NLP foundation models, prompt learning learns contextual text tokens to append and improve the raw questions through various techniques Jiang et al. (2020); Shin et al. (2020); Zhong et al. (2021); Li & Liang (2021); Lester et al. (2021). In CFMs, prompt learning Zhou et al. (2022b); Parisot et al. (2023); Bulat & Tzimiropoulos (2022); He et al. (2022) optimizes text tokens to improve image classification based on image descriptions. Our research is built upon SSprompt Huang et al. (2024), which is the first to introduce prompt learning for SAMs. Different from SSprompt that directly learns semantic/spatial prompts for SAMs, we design a more effective prompt learning method. It explores the idea of Hough voting to learn complementary spatial and semantic prompts that focus on multi-granularity clues with different spatial focus, and vote together to guide SAMs to segment effectively.

**Hough Voting** (Hough, 1959; Ballard, 1981) was widely-used in shape detection and object recognition. It reformulates the detection of patterns as the identification of peaks in a parametric space, which was further extended to allowing for recognize arbitrary shapes. Hough Voting was employed in various tasks including the implicit shape model for object recognition (Leibe et al., 2008), plane extraction from 3D point clouds (Borrmann et al., 2011), and 6D pose estimation (Sun et al., 2010; Kehl et al., 2016). Learning-based extensions of Hough voting also emerged, where weighted votes were introduced to indicate significance in object detection (Maji & Malik, 2009). Hough forests (Gall et al., 2011; Gall & Lempitsky, 2013) combined ensembling with Hough Voting to improve detection accuracy. HoughNet (Samet et al., 2020; 2022) aggregates feature votes within neural network, resulting in precise object detection. Hough voting has also been applied to 3D object recognition (Knopp et al., 2010; Woodford et al., 2014; Velizhev et al., 2012; Knopp et al., 2011; Qi et al., 2019). Inspired by Hough Voting, this work aims to learn complementary spatial prompts and semantic prompts that focus on multi-granularity spatial clues of objects, and vote together to guide SAMs to generate a precise segmentation mask for the objects.

## 3 METHOD

### 3.1 PRELIMINARIES

**Segmentation Anything Models (SAMs)** (Kirillov et al., 2023; Zou et al., 2023; Li et al., 2023) are designed to segment anything by utilizing a "Promptable Segmentation" scheme, where the model generates segmentation masks based on given prompts, such as text, points, coarse masks and images. SAMs can segment any objects and background stuff when appropriate prompts are provided. Additionally, SAMs support "interactive segmentation," which allows users to iteratively refine segmentations and scale up training data through model-in-the-loop annotation. This process more powerful SAMs that can handle diverse segmentation tasks with improved flexibility and accuracy.

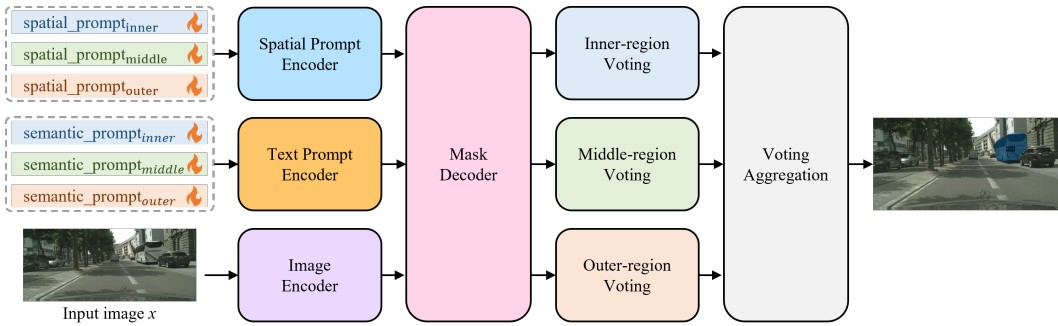

Figure 2: **Framework of HoughPL with HoughSpaPL and HoughSemPL.** HoughSpaPL introduces three types of voting mechanisms including *inner-region voting*, *middle-region voting* and *outer-region voting*, which encourage to learn three spatial prompts with different spatial focuses that provide complementary spatial clues and vote together to generate a robust segmentation mask. Following the same philosophy, HoughSemPL learns distinct semantic prompts for different sub-regions of the visual concept, which capture spatially-complementary semantic clues and vote together to predict a accurate semantic label for the generated mask.

SAM typically works in two stages as illustraced in Figure 3. It first encodes an input image and a set of prompts into feature embeddings and then predicts the expected segmentation mask conditioned on these features. Specifically, SAM consists of an image encoder $\textbf{Encoder}^I$, a text prompt encoder $\textbf{Encoder}^T$, a spatial prompt encoder $\textbf{Encoder}^S$, and a mask decoder $\textbf{Decoder}$. Given an input image $x^I \in \mathbb{R}^{H \times W \times 3}$ and a set of prompts including a spatial prompt $x^S$ and a semantic prompt $x^T$, SAM first encodes $x^I$, $x^S$, and $x^T$ into $D$-dimensional embeddings as follows:

$$z^I = \textbf{Encoder}^I(x^I), \ z^S = \textbf{Encoder}^S(x^S), \ z^T = \textbf{Encoder}^T(x^T), \tag{1}$$

The image feature embeddings are then fed into the to The mask decoder $\textbf{Decoder}$ then decodes the image feature embeddings $z^I$ conditioned on the prompt embeddings $z^S$ and $z^T$:

$$(m, c) = \textbf{Decoder}\left(z^I \mid (z^S, z^T)\right), \tag{2}$$

where $z^S$ and $z^T$ interact with $z^I$, leading to a predicted binary segmentation mask $m$ and the corresponding predicted confidence score $c$.

Under the setting of zero-shot cross-dataset inference scheme, SAM first handle a set of default prompts (i.e., raw category names $X_{\text{default}}^T$ as semantic prompts and a grid of points $X_{\text{default}}^S$ as spatial prompts) by:

$$Z_{\text{default}}^S = \textbf{Encoder}^S(X_{\text{default}}^S), \ Z_{\text{default}}^T = \textbf{Encoder}^T(X_{\text{default}}^T), \tag{3}$$

then predict a set of segmentation masks for $x^I$ as follows

$$(M, C) = \textbf{Decoder}\left(z^I \mid \left(Z_{\text{default}}^S . Z_{\text{default}}^T\right)\right), \tag{4}$$

However, directly using default prompts for downstream datasets is usually sup-optimal, and how to acquire suitable prompts for SAM is a non-trivial task but largely under-explored. This work investigates how to learn effective prompts for SAM with few-shot data.

**Prompt Learning** aims to transfer a foundation model towards downstream domain by optimizing the prompts with a few labeled data. Various prompt learning methods have been proposed for image classification foundation models (CFMs) and segment anything models (SAMs). SAM prompt learning methods aims to learn effective semantic/spatial context tokens to imporve the raw class names and default spatial prompts, for better prompting the text/spatial prompt encoder. Specifically, $M$ learnable context tokens(i.e., $X_{\text{learnt}}^T = \{x_1^T, x_2^T, ..., x_M^T\}$ as text tokens while $X_{\text{learnt}}^S = \{x_1^S, x_2^S, ..., x_M^S\}$ as spatial tokens) are introduced to model the semantic/spatial context of raw class names/default spatial prompts. Given an image $x^I$, the learned prompts $X^T$ and $X^S$, the image segmentation prediction can be formulated by:

$$(M, C) = \textbf{Decoder}\left(z^I \mid \left(Z_{\text{learnt}}^S, Z_{\text{learnt}}^T\right)\right), \tag{5}$$

where $Z_{\text{learnt}}^S = \textbf{Encoder}^S(X_{\text{learnt}}^S)$ and $Z_{\text{learnt}}^T = \textbf{Encoder}^T(X_{\text{learnt}}^T)$. To transfer a SAM towards a downstream domain, an image segmentation loss can be adopted to optimize $X^T$ and $X^S$ over few shot data while keeping the SAM frozen.

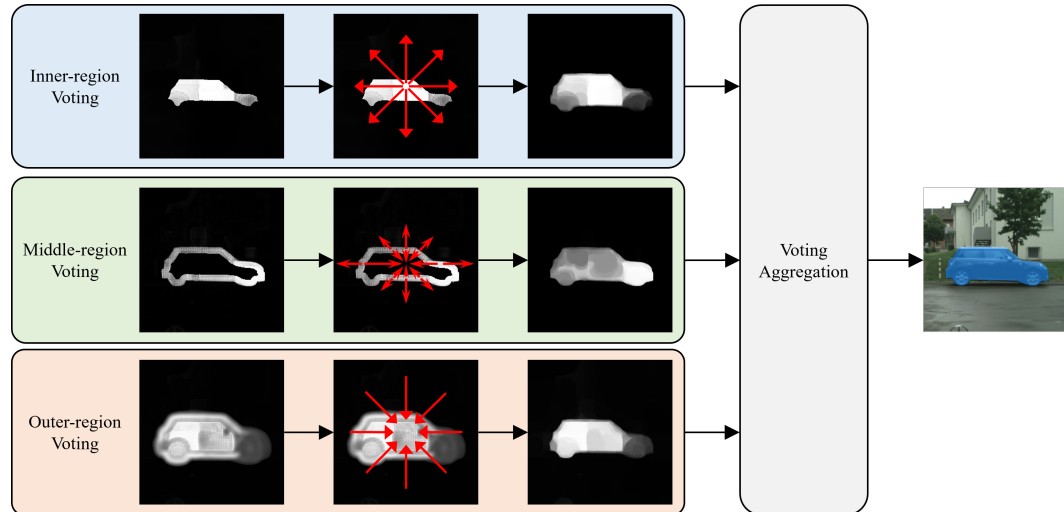

Figure 3: **Illustration of voting mechanisms.** Inner-region voting predicts a segmentation mask for a whole object by voting from the information learnt from the inner-region only, which enforces the inner-region prompt to focus on and learn from the inner-region of object/background. Similarly, middle-region voting enforces the prompt to learn and vote from the middle-region of object (e.g., the region around object boundary) a segmentation mask for a whole object. And outer-region voting enforces the prompt to learn and vote from the outer-region of object (e.g., the region outside object that may contain other objects and/or background) a segmentation mask for a whole object. In this way, the three types of voting mechanisms encourage to learn complementary spatial prompts which together guide SAMs to generate a precise segmentation mask.

Different from previous work that directly learns semantic/spatial prompts for SAMs, we design a more efficient and effective prompt learning method for SAMs. Specifically, it explores the idea of Hough voting to learn complementary spatial and semantic prompts and vote together to guide SAMs to segment effectively.

## 3.2 HOUGH VOTING-BASED PROMPT LEARNING

We focus on prompt learining for SAMs with a few labeled downstream domain data. We identified that one instance can be complementarily described through different views of spatial prompts and their semantic prompts, which focus on regions with various attributes. Inspired by Hough Voting (Ballard, 1981; Woodford et al., 2014; Qi et al., 2019; Samet et al., 2022) that detects a object based on the voting from its parts, we propose to learns distinct spatial prompts for different sub-regions of the object, which capture complementary spatial clues and vote together to guide SAMs to generate a precise segmentation mask for the object. To this end, we design Hough voting based prompt learning (HoughPL) that introduces Hough Voting-based Spatial Prompt Learning (HoughSpaPL) and Hough Voting-base Semantic Prompt Learning (HoughSemPL), as illustraed in Figure 3. The two prompt learning methods complement each other by learn three spatial/semantic prompts with different spatial/semantic focus respectively and vote together to prompt SAMs .

### 3.2.1 HOUGH VOTING-BASED SPATIAL PROMPT LEARNING (HOUGHSPAPL)

HoughSpaPL designs three types of voting mechanisms (i.e., inner-region voting, middle-region voting and outer-region voting) and learns distinct spatial prompts on the embedding space for different sub-regions of the objects, which capture complementary spatial clues and vote together to guide SAMs to generate a precise segmentation mask for the object. In this way, HoughSpaPL learns effective spatial prompts for SAMs with two desirable features: 1) it learns different spatial prompts that capture complementary spatial clues, which can facilitate locating the target thing/stuff and predicting a precise segmentation boundary, leading to a robust segmentation; 2) its voting mechanisms enable and encourage information flow among different regions of the whole object, learning rich contextual information and benefiting image segmentation.

Let $Z_{\text{inner}}^S$, $Z_{\text{middle}}^S$, and $Z_{\text{outer}}^S$ denote three types of spatial prompts with $N$ learnable embeddings that capture complementary spatial clues as follows,

$$Z_{\text{inner}}^S = \left\{ z_{\text{inner}}^S(n) \right\}_{n=1}^N, \ Z_{\text{middle}}^S = \left\{ z_{\text{middle}}^S(n) \right\}_{n=1}^N, \ Z_{\text{outer}}^S = \left\{ z_{\text{outer}}^S(n) \right\}_{n=1}^N, \tag{6}$$

where $Z_{\text{inner}}^S$ denotes inner-region spatial embeddings that capture the spatial clues near object center region, which predicts a segmentation mask for a whole object by voting from the information learnt from the inner-region only, $Z_{\text{middle}}^S$ denotes middle-region spatial embeddings that capture the spatial clues and vote a segmentation mask from the information learnt near around the object boundary, $Z_{\text{outer}}^S$ denotes outer-region spatial embeddings that capture the spatial clues and vote a segmentation mask from the information learnt from outside object that may contain other objects and/or background.

Given an image $x^I$ and the spatial prompts (i.e., $Z_{\text{inner}}^S$, $Z_{\text{middle}}^S$, and $Z_{\text{outer}}^S$), SAMs predict a set of segmentation masks:

$$\{(M_i, C) \mid i \in \mathcal{H}\} = \textbf{Decoder}\left(z^I \mid \left\{ \left(Z_i^S, Z_{\text{default}}^T\right) \mid i \in \mathcal{H} \right\}\right), \tag{7}$$

where $\mathcal{H} = \{\text{inner}, \text{middle}, \text{outter}\}$. HoughSpaPL aggregate three segmentation masks (i.e., $\{M_i \mid i \in \mathcal{H}\}$) to derive a final prediction $M$ that aggregates the information from these predictions, works as described below.

**Inner-region Voting** enforces the prompt to focus on and learn from the inner-region of object It predicts a segmentation mask for a whole object by voting from the information learnt from the inner-region only. Specifically, for the pixel at $(i, j)$, its output of inner-region voting is set to the maximum gray value found in its neighborhood, which can be formulated as:

$$\tilde{M}_{\text{inner}}(i, j) = \max_{(p_i, p_j) \in \mathcal{N}_r(i,j)} M_{\text{inner}}(i + p_i, j + p_j) \tag{8}$$

where $\mathcal{N}_r(i, j)$ is the neighborhood with size $r$ around pixel $(i, j)$, $(p_i, p_j)$ is the offset within the neighborhood. This voting process focuses on expanding the inner regions of the object. The center of the object grows outward by selecting the highest intensity values in the neighborhood, ensuring that the most confident areas of the object dominate the segmentation. The inner-region provides a strong spatial clues by capturing the core of the object.

**Middle-region Voting** focuses on the boundary of the object, ensuring that the edges are well defined. It involves two steps: first, expand boundary region by considering the brightest surrounding pixels, followed by suppress the over expanded region to refines the object boundary, ensuring the boundaries are smooth and intact. Specifically, the output of middle-region voting at $(i, j)$ is set as,

$$\begin{aligned} \hat{M}_{\text{middle}}(i, j) &= \max_{(p_i, p_j) \in \mathcal{N}_r(i,j)} M_{\text{middle}}(i + p_i, j + p_j), \\ \tilde{M}_{\text{middle}}(i, j) &= \min_{(q_i, q_j) \in \mathcal{N}_r(i,j)} \hat{M}_{\text{middle}}(i + q_i, j + q_j). \end{aligned} \tag{9}$$

This voting process refines the middle regions of the object, particularly near the boundaries. It ensures that any small holes or gaps in the object boundaries are filled while preserving the object's shape. The middle-region voting enforces the segmentation mask to focus on these boundaries areas, ensuring the boundaries are well-defined.

**Outer-region Voting** emphasizes voting from regions outside the object, helping to remove background or unrelated objects. This is done by selecting the minimum value from the local neighborhood, which shrinks the outer region as follows,

$$\tilde{M}_{\text{outer}}(i, j) = \min_{(p_i, p_j) \in \mathcal{N}_r(i,j)} M_{\text{outer}}(i + p_i, j + p_j). \tag{10}$$

The outer-region voting encourages the segmentation process to eliminate weakly connected or irrelevant background around the object. By choosing the minimum values from the neighborhood, it effectively shrinks the segmentation mask, removing noise and background elements, and ensuring the segmentation mask is focused on the core of the object without interference from external regions.

**Voting Aggregation:** The final segmentation mask is computed as the aggregation of the results predicted by above three voting mechanisms as follows,

$$M_{\text{final}} = \text{Average}(\tilde{M}_{\text{inner}}, \tilde{M}_{\text{middle}}, \tilde{M}_{\text{outer}}), \tag{11}$$

where we can employ a segmentation loss to optimize $\{Z_i^S \mid i \in \mathcal{H}\}$ to find the best spatial prompts that capture complementary spatial clues for SAMs with respect to each downstream dataset. Note, during training, we only update the learnable embeddings $\{Z_i^S \mid i \in \mathcal{H}\}$, while all other modules are frozen as illustrated in Figure 3.

### 3.2.2 Hough Voting-based Semantic Prompt Learning (HoughSemPL)

HoughSemPL learns three types of semantic prompts associated with the three spatial prompts in HoughSpaPL respectively. Specifically, inner-region semantic prompt learns to recognize the object category based on the inner-region features, middle-region semantic prompt focuses on boundary semantics to enhance the object classification, outer-region semantic prompt facilitates distinguishing the object from the background by learning the surrounding context. By combining these spatial clues coupled semantic prompts, HoughSemPL achieves a precise and robuts segmentation mask that incorporates both spatial structure and semantic information. In this way, HoughSemPL learns effective semantic prompts for SAMs with two desirable features: 1) it learns different semantic prompts that captures multi-granularity semantics by extracting spatial clues from the inner, middle, and outer regions of objects, facilitating precise object segmentation and recognition. 2) its voting mechanisms combines multi-granularity semantics to develop a comprehensive understanding of the object context, effectively handling object occlusions and background interference, resulting in robust segmentation.

Given an image $x^I$ and learnable semantic prompts (i.e., $Z_{\text{inner}}^T$, $Z_{\text{middle}}^T$, and $Z_{\text{outer}}^T$) and learnable spatial prompts in Eq. (6), SAMs predict a set of segmentation masks:

$$\{(M_i, C_i) \mid i \in \mathcal{H}\} = \textbf{Decoder}\left(z^I \mid \left\{\left(Z_i^S, Z_i^T\right) \mid i \in \mathcal{H}\right\}\right), \tag{12}$$

where $\mathcal{H} = \{\text{inner}, \text{middle}, \text{outter}\}$. HoughSemPL aggregate and votes the predictions from the three segmentation mask classification (i.e., $\{C_i \mid i \in \mathcal{H}\}$) to derive a final prediction $C_{\text{final}}$ as follows,

$$C_{\text{final}} = \text{Average}(C_{\text{inner}}, C_{\text{middle}}, C_{\text{outer}}). \tag{13}$$

Note that although HoughSemPL are simple in form as Eq. (12) and (13), the learnable semantic prompts are tightly coupled with the learnable spatial prompts, which enables the effective extraction of multi-granularity semantic information that focus on spatial clues from different regions of object: core features from the inner region, boundary information from the middle region, and background context from the outer region, respectively This coupling of spatial and semantic prompts leads to an accurate and robust object segmentation. We can employ a segmentation loss to optimize both the learnable spatial prompts (i.e., $\{Z_i^S \mid i \in \mathcal{H}\}$) and learnable semantic prompts (i.e., $\{Z_i^T \mid i \in \mathcal{H}\}$) to find the best spatial and semantic prompts that focus on complementary spatial clues for SAMs with respect to each downstream dataset. Note, during training, we only update the learnable embeddings $\{(Z_i^S, Z_i^T) \mid i \in \mathcal{H}\}$, while all other modules are frozen as illustrated in Figure 3.

## 4 Experiments

Tables 1-2 show the benchmarking of our methods with state-of-the-art prompt learing methods, including CoOp (Zhou et al., 2022b), LOCN (Parisot et al., 2023), and SSPrompt (Huang et al., 2024), over 4 widely used image segmentation datasets inclduing Cityscapes (Cordts et al., 2016), Mapillary (Neuhold et al., 2017), ADE20K (Zhou et al., 2017) and ACDC (Sakaridis et al., 2021). The 4 dataset have diverse scene and rich semantic categories captured in different cities under various illumination conditions and weather conditions. We did not include COCO dataset in experiments as it has been used in SAMs pre-training.

### 4.1 Implementation Details

We conduct experiments on pre-trained segment anything model SEEM (Zou et al., 2023) with vision backbone Focal-Tiny (Yang et al., 2022) and Davit-Large (Ding et al., 2022). We adopt SGD optimizer with an initial learning rate of $1 \times 10^{-3}$, a weight decay of $1 \times 10^{-4}$, and a polynomial learning rate schedule with a power of $0.9$. The input images are resized, where the shorter side is set to $512$ pixels, and random flipping is employed as a data augmentation. The number of semantic prompts $C$ is determined by the number of categories in downstream dataset. Following

Table 1: Prompt learning of segmentation anything models on common datasets. The experiments are conducted on semantic segmentation (in mIoU), instance segmentation (AP50), and panoptic segmentation (PQ), where 16-shot data are used (i.e., 16 labelled images for each class) for each dataset.

| Method | Cityscapes | | | Mapillary | | | ADE20K | | |
|---|---|---|---|---|---|---|---|---|---|
| | Sem. Seg | Ins. Seg | Pan. Seg | Sem. Seg | Ins. Seg | Pan. Seg | Sem. Seg | Ins. Seg | Pan. Seg |
| SEEM-T (Zou et al., 2023)(Baseline) | 39.2 | 32.7 | 32.4 | 42.1 | - | 32.7 | 14.6 | 10.2 | 10.5 |
| CoOp (Zhou et al., 2022b) | 50.1 | 34.7 | 35.9 | 43.3 | - | 33.9 | 17.6 | 11.9 | 11.6 |
| LOCN (Parisot et al., 2023) | 51.5 | 35.6 | 36.3 | 44.2 | - | 34.2 | 19.3 | 12.3 | 12.7 |
| SSPrompt (Huang et al., 2024) | 55.2 | 37.7 | 38.1 | 49.5 | - | 36.5 | 23.2 | 15.1 | 15.0 |
| HoughPL (Ours) | 57.5 | 40.1 | 40.5 | 52.3 | - | 38.1 | 25.6 | 18.7 | 19.8 |

Table 2: Prompt learning of segmentation anything models on adverse-condition dataset ACDC (Sakaridis et al., 2021). The experiments are conducted on semantic segmentation (in mIoU), instance segmentation (in AP50), and panoptic segmentation (in PQ), where 16-shot data are used (i.e., 16 labelled images for each class) for each condition.

| Method | Foggy Condition | | | Night Condition | | | Rain Condition | | | Snow Condition | | |
|---|---|---|---|---|---|---|---|---|---|---|---|---|
| | Sem. Seg | Ins. Seg | Pan. Seg | Sem. Seg | Ins. Seg | Pan. Seg | Sem. Seg | Ins. Seg | Pan. Seg | Sem. Seg | Ins. Seg | Pan. Seg |
| SEEM-T (Zou et al., 2023) (Baseline) | 34.6 | 35.5 | 29.3 | 26.2 | 12.8 | 15.3 | 33.1 | 19.8 | 22.6 | 35.8 | 27.5 | 23.5 |
| CoOp (Zhou et al., 2022b) | 36.7 | 36.6 | 31.2 | 28.6 | 16.1 | 17.2 | 33.5 | 25.0 | 27.7 | 36.4 | 29.1 | 28.4 |
| LOCN (Parisot et al., 2023) | 40.1 | 38.0 | 32.1 | 29.1 | 17.0 | 17.9 | 34.1 | 25.3 | 28.1 | 36.6 | 29.9 | 28.4 |
| SSPrompt (Huang et al., 2024) | 47.5 | 40.8 | 35.2 | 32.1 | 19.6 | 20.0 | 39.9 | 29.9 | 32.6 | 43.1 | 32.6 | 31.7 |
| HoughPL (Ours) | 56.4 | 44.8 | 40.5 | 35.2 | 21.9 | 21.8 | 42.4 | 31.3 | 36.8 | 47.5 | 33.9 | 34.8 |

the configuration in Zou et al. (2023), we set the number of spatial prompts $N$ and the dimension of embedding $D$ as 100 and 512 respectively, and employ multi-category cross-entropy loss for class prediction training and binary cross-entropy loss for mask prediction training.

## 4.2 RESULTS

**Prompt Learning for SAM on Common Datasets.** Table 1 reports the image segmentation results on three widely-used common datasets: Cityscapes, Mapillary Vistas, and ADE20K. HoughPL demonstrates significant performance gains over the baseline across these datasets. Specifically, HoughPL outperforms the state-of-the-art methods by 5.5 in mIoU for semantic segmentation, 5.5 in AP for instance segmentation, and 5.5 in PQ for panoptic segmentation on average. This highlights the superiority of HoughPL in effectively transferring pretrained SAM to various segmentation dataset with limited labeled dataset. The superior performance of HoughPL largely attributed to its ability of capturing multi-granularity spatial and semantic information, which allows it to focus on multi-granularity spatial clues from the inner, middle, and outer regions of objects. Additionally, HoughPL facilitates comprehensive contextual understanding, enabling it to adapt flexibly to diverse segmentation scenarios across datasets. All compared methods improve the performance while HoughPL achieves the most substantial perfomance gains, demonstrating that HoughPL can effectively align the pretrained model's representation with the various segmentation tasks by efficiently learning complementary spatial and semantic prompts from multi-granularity spatial and semantic clues. Note that due to the inefficiency of converting the high-resolution Mapillary Vistas dataset to COCO-style instance annotations, we only report the results for panoptic and semantic segmentation on this dataset.

**Prompt Learning for SAM on Adverse-Condition Datasets.** Table 2 reports the experimental results on autonomous driving datasets under various weather and time conditions. Notably, HoughPL consistently surpasses all other methods by significant margins, which demonstrate its effectiveness and robustness in adapting SAM to adverse conditions with considerable variations in lighting and scene content. The performance gains achieved by HoughPL in adverse conditions highlight the effectiveness of the designed voting mechanisms as well as learning complementary spatial and semantic prompts with multi-granularity spatial focus from limited labeled samples.

## 4.3 DISCUSSION

**Generalization across different datasets.** We evaluate the generalization of HoughPL through extensive experiments on different segmentation tasks over multiple datasets. The experimental results in Tables 1 and 2 demonstrate that HoughPL consistently outperforms other methods on 4

Table 3: Performance versus number of data on semantic segmentation (in mIoU) over Cityscapes.. The default is marked in gray .

| SEEM-T (Baseline) | HoughPL | | | |
| | 16-shot | 12-shot | 8-shot | 4-shot |
|---|---|---|---|---|
| 39.2 | 57.5 | 54.3 | 51.6 | 49.5 |
| SEEM-L (Baseline) | 16-shot | 12-shot | 8-shot | 4-shot |
| 49.3 | 60.4 | 57.9 | 56.0 | 54.4 |

Table 4: Ablation studies of HoughPL with HoughSpaPL and HoughSemPL. The experiments are conducted on semantic segmentation (in mIoU) over Cityscapes.

| Method | Hough voting-based Spatial Prompt Learning | | | HoughSemPL | mIoU |
| | Inner-region Voting | Middle-region Voting | Outer-region Voting | | |
|---|---|---|---|---|---|
| SEEM-T (Baseline) | | | | | 39.2 |
| | ✓ | | | | 40.4 |
| | ✓ | ✓ | | | 44.9 |
| | ✓ | ✓ | ✓ | | 48.7 |
| | | | | ✓ | 52.6 |
| **HoughPL (Ours)** | ✓ | ✓ | ✓ | ✓ | 57.4 |

widely studied datasets, including both common and adverse-condition scenarios, highlighting its effectiveness and robustness of in handling diverse segmentation tasks and datasets.

**Generalization across different vision backbones.** We conduct experiments to verify the generalization of HoughPL by evaluating it with two vision backbones: Focal-Tiny (Yang et al., 2022) and Davit-Large (Ding et al., 2022). As Tables 3 demonstrates, HoughPL performs well consistently across both small and large backbones, verifying its robustness in diverse architectural settings.

**Ablation study.** We conduct ablation study on Cityscapes dataset to analyze the contribution of the components in HoughPL. As shown in Table 4, the baseline does not perform well, while incorporating Inner-region Voting improves the baseline, indicating that this mechanism helps focus on inner regions of objects, enabling better localization. In addition, the Middle-region Voting further improves performance, demonstrating that the boundary information in middle-region effectively regularizes the learning process of spatial prompts. Additionally, Outer-region Voting leads to further performance improvement, largely attribute to its captured contextual information that enhances comprehensive spatial understanding, demonstrating that the combination of the above three voting mechanism regularizes the learning process more holistically. Finally, the combination of HoughSpaPL and HoughSemPL achieves the best results, demonstrating that the complementary spatial and semantic prompts that focus on spatial cues at different levels of granularity complement each other effectively, leading to robust and effective segmentation, highlighting the advantage of unifying multi-granularity spatial and semantic information.

**Performance versus the number of training data.** We assess the generalization of HoughPL by conducting evaluations across various amounts of training data ranging from 16 shot to 4 shot on the Cityscapes dataset. The experimental results in Tables 3 show that HoughPL consistently achieves superior performance across different amount of training data, highlighting the its effectiveness in handling different scales of training samples, validating its efficiency in data-scarce scenarios.

**Efficiency comparison.** We study the training efficiency of all the compared methods. Table 5 reports the results on ADE20K dataset, which indicate that HoughPL with multiple prompts and voting mechanism achieves comparable training efficiency to CoOp and LOCN, demonstrating that our designs does not significantly influence the training efficiency while achieve superior performance.

Table 5: Training and inference efficiency comparison in time (ms per image).

| SEEM-T (Baseline) | CoOp | LOCN | SSprompt | HoughPL (Ours) |
|---|---|---|---|---|
| Training Time | 96.4 | 98.2 | 60.5 | 62.4 |
| Inferece Time | 24.1 | 23.5 | 25.8 | 29.2 |

**Parameter analysis.** In HoughSpaPL, the neighborhood size of the voting region is determined by a pre-defined scalar $r$. We studied the impact of varying $r$ by changing it from 3 to 11 with a step of 2. Table 6 reports the experimental results on Cityscapes dataset. Results indicate that a smaller $r$ limits the ability of capturing broader contextual spatial clues, while a larger $r$ causes excessive smoothing, which failes to perserve the boundary details. With an appropriately $r$, HoughPL achieves a balance between capturing fine-grained local information and maintaining a comprehensive and complementary spatial and semantic clues of objects, leading to optimal segmentation performance.

Table 6: Parameter analysis of HoughPL for the voting region size $r$ on semantic segmentation (in mIoU) over Cityscapes.

| $r$ | 3 | 5 | 7 | 9 | 11 |
|---|---|---|---|---|---|
| mIoU | 53.8 | 56.6 | 57.5 | 57.3 | 57.0 |

**HoughPL versus Other Voting methods.** For fair comparison, we compared our HoughPL with Vanilla Voting that simply average three spatial and three semantic prompts respectively. Results in Table 7 show that Vanilla Voting does not help much, largely because that simply averaging three learnt spatial and three learnt semantic prompts respectively lacks appropriate training regularization, leading to failure of capturing the multi-granularity spatial clues with different spatial focus. On the other hand, our HoughPL effectively learns the spatial and semantic prompts that focus on multi-granularity spatial clues of objects, leading to superior segmentation performance.

Table 7: Comparing our HoughPL to Vanilla Voting on semantic segmentation over Cityscapes.

| Method | SEEM-T (Baseline) | Vanilla Voting | HoughPL (Ours) |
|---|---|---|---|
| mIoU | 39.2 | 50.7 | 57.5 |

## 5 CONCLUSION

This paper presents HoughPL, a novel prompt learning framework that employs voting mechanisms to adapt SAMs towards various downstream segmentation dataset. By designing inner-region voting mechanisms, middle-region voting mechanisms, and outer-region voting mechanisms that capture multi-granularity spatial cues, HoughPL learns multiple complementary spatial and semantic prompts and votes them together to guide SAMs for more effective segmentation. By effectively guiding SAMs to adapt to downstream datasets with limited labeled data, HoughPL significantly improves segmentation performance in both common and challenging adverse-condition scenarios. Extensive experiments across multiple widely-adopted segmentation datasets demonstrate that HoughPL consistently outperforms state-of-the-art methods by clear margins.

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

## A  APPENDIX

You may include other additional sections here.

