# OpenReview forum: "Hough Voting-based Prompt Learning for Segment Anything Model"
_ICLR.cc/2025/Conference — ICLR 2025 Conference Withdrawn Submission_

### Official Review · Reviewer_Fi7w · 2024-10-25

**Soundness:** 2
**Presentation:** 3
**Contribution:** 1
**Rating:** 3
**Confidence:** 5

**Summary:**

The paper introduces a novel approach to prompt learning for Segment Anything Models (SAMs), which are designed to segment images into various objects and background regions. The authors propose Hough Voting-based Spatial Prompt Learning (HoughSpaPL) and Hough Voting-based Semantic Prompt Learning (HoughSemPL), which leverage the concept of Hough Voting to learn distinct spatial and semantic prompts for different subregions of visual concepts. These prompts capture complementary spatial and semantic clues that vote together to guide SAMs in generating precise segmentation masks. The authors claim that their method, HoughPL, outperforms existing prompt learning techniques on multiple segmentation datasets and under various adverse conditions.

**Strengths:**

1、The paper presents a creative application of Hough Voting for prompt learning in SAMs, which is a novel concept in the field of image segmentation.

2、 The authors have conducted extensive experiments across multiple datasets, demonstrating the effectiveness of their method in various scenarios, including adverse conditions.

3、The paper claims significant performance improvements over existing methods, which is a strong point in favor of the proposed technique.

**Weaknesses:**

1、Providing more precise prompts to models like SAM is a topic worth considering and exploring, but the inter, middle, and outer prompts  proposed in this paper (as shown in Figure 1) can actually be solved by adding points to the original SAM model. In other words, the comparison in Figure 1 itself is unfair, because the results of one prompt cannot be compared with the results of three prompts, and if the number of prompt in the first row is increased to 3 (the original SAM has this capability), the effect should not be worse than that of the second row.

2、 Based on the first point, the comparison of the effects of these two methods is missing, so it seems that the results of this paper are hard to be convincing. In addition, there is no comparison of the effects of the SAM and SAM2 models themselves, as well as some derivative versions of the SAM model, such as mobilesam, efficientsam, etc.

3、The proposed method may be more complex than necessary, introducing additional mechanisms that might not significantly improve over simpler models.

4、 The paper does not discuss the computational efficiency of the proposed method, which could be a concern for practical applications.

5、 The experiments are limited to a specific set of datasets, and the performance on other types of datasets or real-world scenarios is not discussed.

6、 How does HoughPL compare with the latest state-of-the-art methods in terms of accuracy, robustness, and efficiency?
Scalability: Can the proposed method scale to larger and more complex datasets?

**Questions:**

Refer to Weaknesses part.

---

### Official Review · Reviewer_qw9Y · 2024-10-31

**Soundness:** 3
**Presentation:** 2
**Contribution:** 3
**Rating:** 5
**Confidence:** 4

**Summary:**

This paper proposes a Hough voting-based prompt learning method for segment anything models to make them more effective in different downstream tasks. The proposed method consists of a Hough Voting-based Spatial Prompt Learning (HoughSpaPL) module and a Hough Voting-based Semantic Prompt Learning (HoughSemPL) module. HoughSpaPL generates the precise segmentation mask by a voting machanism from inner region, middle region and outer region. HoughSemPL learns the semantic prompt from the three different region following the same philosophy. Experiments show the proposed method achieve a superior performance over other SOTA prompt learning methods for SAMs.

**Strengths:**

1. Simple but effective method. The proposed method is simple but shows a superior performance over exsiting methods.
2. The idea of introducing Hough voting in prompt learning is somewhat novel.

**Weaknesses:**

1. Although Hough voting is novel in promt learning, the whole structure is an incremental change by deleting the default text and spatial prompts based on SSprompt [Huang et al. 2024] method.
2. Unclear method implementation. Where is the segmentation loss applied during training? Before region voting or after region voting? In my understanding, the semantic prompt number C is much smaller than spatial prompt number N. How are semantic prompts assigned to different spatial region for capturing multi-granularity semantics by extracting spatial clues from the inner, middle, and outer regions of objects.
3. As the spatial promt number for each region is N, there are 3N spatial prompts in total, which is three times of the SSprompt. This is unfair in the comparison. Authors should have some experiments on the influence of prompt number.
4. In line 407, the authors claim that HoughPL outperforms the SOTA methods with 5.5 in mIoU, 5.5 in AP and 5.5 in PQ on average. But table 1 does not support it.
5. Lots of grammar errors and typos. For example, line 23: "a accurate" -> "an accurate". line051: "to learns" -> "to learn". line156: "Segmentation Anything Models (SAMs)" -> "Segment Anything Models (SAMs). line187: " ...fed into the to The mask deocder". line289: "object It". and ect.

**Questions:**

1. In table 4, it shows the semantic prompt plays the most important role in performance improvment. How is Hough semantic prompt learning implemented without region voting? This paper aims to solve the problem of learning spatial prompts. Whether the results support the contribution that authors claimed needs more explanation.

---

### Official Review · Reviewer_mw5w · 2024-11-02

**Soundness:** 3
**Presentation:** 2
**Contribution:** 3
**Rating:** 5
**Confidence:** 4

**Summary:**

The paper introduces HoughPL, a novel mechanism for learning both spatial and semantic prompts for Segment Anything Models (SAMs). The proposed approach employs a voting mechanism where prompts with different focus areas collaborate to achieve competitive performance on standard benchmarks.

**Strengths:**

1. The paper presents its ideas with clarity and good technical writing.
2. The empirical evaluation is thorough; well-designed ablation studies demonstrate each component's contribution.
3. The multi-focus spatial prompt approach aligns well with the inherent nature of segmentation tasks

**Weaknesses:**

1. Despite frequent references to "SAMs" throughout the paper, the experimental validation is conducted exclusively on SEEM, which has lower adoption than SAM. This raises questions about the approach's generalization capabilities across other SAM variants like the original SAM and Semantic-SAM mentioned in the paper.
2. The manuscript contains editorial oversights (e.g., incorrect figure reference on Line 180, which should be "Figure 2").
3. The visual evidence provided is somewhat limited:
    - Figure 1's nature (whether it's a schematic diagram or actual learnable prompt output) needs clarification
    - The paper would benefit from additional visualizations of learnable prompt results and their corresponding mask outputs
    - The current visual examples are limited to four images, all from traffic scenes, limiting domain diversity

**Questions:**

1. Could you elaborate on the "multi-granularity" aspect of semantic prompts? Specifically:
    - How does this approach differ from SAM's existing three learnable tokens?
    - What advantages does your multi-granularity implementation offer over SAM's built-in multi-granularity handling through self-attention with input spatial embedding?
2. The ablation studies suggest that HoughSemPL contributes the majority of the performance improvements. Could you provide more detailed insights into this component's effectiveness?

---

### Official Review · Reviewer_TJSH · 2024-11-02

**Soundness:** 2
**Presentation:** 2
**Contribution:** 2
**Rating:** 3
**Confidence:** 4

**Summary:**

The paper proposes Hough Voting-based Spatial Prompt Learning (HoughSpaPL) for the segment anything model. Three kinds of spatial prompts correspond to inner-region, middle-region and outer-region votings are designed to generate a final segmentation task.  HoughSemPL is also introduced to complement HoughSpaPL with semantic cues. The experiments are validated on Cityscapes, Mapillary,  ADE20K and ACDC datasets.

**Strengths:**

1. In Table 4, the paper has a clear ablation experiment to validate the performance boost brought by each proposed component. This shows the gain brought by HoughSpaPL and HoughSemPL.

2. Voting-based prompt learning is an interesting idea to explore, where promting using complementary spatial prompts and
semantic prompts for voting aggregation is an reasonable design.

3. Figure 3 is a good illustration on the roles played by different prompts, making the paper easier to understand. For outer-region voting, when the outer region contain other objects, the designed algorithm still enforces it to predict the target object segmentation mask, why does inner-and-middle not sufficient enough?

**Weaknesses:**

1. It is not clear how does the model handle the dense objects in image cases, where multiple objects of the same category inside the same image. Is the proposed method automatically generate prompt for each object? The paper is not written clear in this part. Also, there are very few visual comparison of semantic/instance-segmentation results in the paper.

2.  Why do separate objects into inside region, middle region and outer region for voting is an optimal design choice? For example, one can seperate an object into more 4 sub-parts like upper-left, upper-right, down-left and down-right. Will more detailed region partition bring more performance gain? There is no enough experiments to validate the core design contribution of the proposed voting schemes and make the paper contribution unclear.

3. There is no comparison to other light-weight / efficient SAM tuning strategy like using LORA or BOFT [a], finetuning partial network layers or adding tokens tuning (like hq-sam) for tuning.

[a] PARAMETER-EFFICIENT ORTHOGONAL FINETUNING VIA BUTTERFLY FACTORIZATION. ICLR, 2024.

4. Could Table 5 of paper also include the comparison on training memory and model parameters sizes?

5. The paper should compare to using default points/box/mask prompts for downstream datasets for segmentation, for example, tuning a prompt encoder to predict bounding boxes or points for SAM's mask decoder as input.

**Questions:**

1. In teaser figure, each sub-figure in the 2nd row has 3 red dot points, are these explicit points directly the output by the HoughSpaPL model?

2. How does the method handle semantic segmentation cases, where regions in the same semantic catergory but disconnected in the image?

---

### Note · Authors · 2024-11-14

I have read and agree with the venue's withdrawal policy on behalf of myself and my co-authors.